# Clinical Study of Metabolic Parameters, Leptin and the SGLT2 Inhibitor Empagliflozin among Patients with Obesity and Type 2 Diabetes

**DOI:** 10.3390/ijms24054405

**Published:** 2023-02-23

**Authors:** Zsolt Szekeres, Barbara Sandor, Zita Bognar, Fadi H. J. Ramadan, Anita Palfi, Beata Bodis, Kalman Toth, Eszter Szabados

**Affiliations:** 1Division of Preventive Cardiology and Rehabilitation, 1st Department of Medicine, Medical School, University of Pecs, 7624 Pecs, Hungary; 2Department of Biochemistry and Medical Chemistry, University of Pecs, Medical School, 7624 Pecs, Hungary; 3Division of Endocrinology and Metabolism, 1st Department of Medicine, Medical School, University of Pecs, 7624 Pecs, Hungary; 4Division of Cardiology, 1st Department of Medicine, Medical School, University of Pecs, 7624 Pecs, Hungary

**Keywords:** empagliflozin, type 2 diabetes mellitus, obesity, leptin, lipid metabolism

## Abstract

Obesity is a major public health problem worldwide, and it is associated with many diseases and abnormalities, most importantly, type 2 diabetes. The visceral adipose tissue produces an immense variety of adipokines. Leptin is the first identified adipokine which plays a crucial role in the regulation of food intake and metabolism. Sodium glucose co-transport 2 inhibitors are potent antihyperglycemic drugs with various beneficial systemic effects. We aimed to investigate the metabolic state and leptin level among patients with obesity and type 2 diabetes mellitus, and the effect of empagliflozin upon these parameters. We recruited 102 patients into our clinical study, then we performed anthropometric, laboratory, and immunoassay tests. Body mass index, body fat, visceral fat, urea nitrogen, creatinine, and leptin levels were significantly lower in the empagliflozin treated group when compared to obese and diabetic patients receiving conventional antidiabetic treatments. Interestingly, leptin was increased not only among obese patients but in type 2 diabetic patients as well. Body mass index, body fat, and visceral fat percentages were lower, and renal function was preserved in patients receiving empagliflozin treatment. In addition to the known beneficial effects of empagliflozin regarding the cardio-metabolic and renal systems, it may also influence leptin resistance.

## 1. Introduction

According to the latest data, nearly 2 billion adults (39% of the world’s adult population) were estimated to be obese or overweight. If current trends continue, it is expected that 1 billion adults, nearly 20% of the world’s population, will clinically be declared obese by 2025 [1]. Obesity is associated with many diseases and abnormalities, such as type 2 diabetes [2], dyslipidemia [3], cardiovascular diseases [4], hypertension [5], certain types of cancer [6,7], pneumological [6], nephrological [8], skeletal muscle [9], rheumatologic [10], dermatologic [11], and neuropsychologic [11] complications, and is it associated with premature mortality. Obesity, especially the dysfunctional visceral adipose tissue (VAT), is the main driver of many metabolic abnormalities including insulin resistance, hyperinsulinemia, glucose intolerance, atherogenic dyslipidemia (high triglyceride and apolipoprotein B levels, increased proportion of small, dense LDL [low-density lipoprotein] particles, low HDL [high-density lipoprotein] cholesterol levels, and small HDL particles), and is associated with a low-grade inflammation [6].

Leptin was the first identified adipokine in the 1990s known to suppress food intake through the suppression of appetite and mediate energy homeostasis including glucose and lipid metabolism [12]. The serum level of leptin is elevated paradoxically in obesity [13], and this high level of leptin may induce leptin resistance and result in altered glucose metabolism and insulin resistance [14]. Hyperleptinemia has also been associated with increased inflammation, oxidative stress, endothelial dysfunction, atherogenesis, and thrombosis [15]. Based on these effects, leptin is attributed to a significant role in the development of cardiovascular diseases. Additionally, patients with type 2 diabetes mellitus scored a higher percentage of hypertension, obesity, metabolic syndrome, and endothelial dysfunction if they had elevated leptin levels [16].

The link between obesity and type 2 diabetes mellitus [T2DM] has long been recognized and explains the high prevalence of type 2 diabetes mellitus. Type 2 diabetes mellitus is associated with many vascular complications. Microvascular complications include diabetic kidney disease, retinopathy, and neuropathy, whereas the macrovascular complications include coronary artery, cerebrovascular, and peripheral vascular diseases. The main goals of treatment in patients with T2DM are to achieve adequate glycemic control, reduce body weight and prevent vascular damage, and target organ damage [17]. Novel antidiabetic therapies such as sodium glucose co-transporter 2 (SGLT2) inhibitors provide a new approach to preventing or ameliorating the complications that insulin resistance and hyperglycemia create [18]. SGLT2 inhibitors are potent antihyperglycemic drugs, which inhibit glucose reabsorption in the proximal tubules of the kidney inducing glycosuria and improving blood glucose levels, and may reduce body weight through calorie loss. Numerous studies have shown they are associated with reduced cardiovascular morbidity and mortality, including vascular diseases and heart failure [19]. Furthermore, SGLT2 inhibitors have also demonstrated positive reno-metabolic effects [20]. In a cardiovascular outcome trial, the SGLT2 inhibitor empagliflozin proved superior to conventional antidiabetic therapy in reducing the rate of MACE, mortality, and hospitalization due to heart failure [21]. SGLT2 inhibitor therapy has been associated with a decrease in serum triglycerides, an increase in HDL cholesterol, and also a small increase in LDL cholesterol level was observed [20]. The presence of metabolic disturbances in obese patients results in oxidative stress [22]. Since obesity and insulin resistance is a major component of metabolic syndrome, it is strongly associated with oxidative stress [23]. The oxidative modification of lipoproteins can result in more atherogenic compounds, which may have a key role in the development of cardiovascular dysfunction in patients with diabetes mellitus [24,25].

The aim of our study was to investigate certain laboratory parameters such as lipids, inflammatory markers, blood glucose level, glycated hemoglobin [HbA1c] level, kidney function, leptin level, as well as body mass index [BMI], body fat and visceral fat percentage among patients afflicted with obesity and diabetes. We also investigated a subgroup of patients receiving empagliflozin treatment.

## 2. Results

### 2.1. Body Mass Index, Body Fat, and Visceral Fat Were Significantly Lower in the Empagliflozin Treated Group

BMI was significantly lower in the control group (C) when compared to the obese (O) (*p* < 0.001), to the obese and diabetic (OD) (*p* < 0.001), and to the empagliflozin treated (ODE) group (*p* < 0.001). It was also significantly lower in the diabetic (D) group when compared to the obese (O) (*p* < 0.001), to the obese and diabetic (OD) (*p* < 0.001), and to the empagliflozin-treated group (*p* < 0.001). BMI was significantly lower in the empagliflozin-treated group (ODE) when compared to the obese and diabetic (OD) group (*p* < 0.001). There was no significant difference between the other groups.

Body fat was significantly lower in the control group (C), when compared to the obese (O) (*p* < 0.001), and to the obese and diabetic (OD) (*p* < 0.001) groups. It was also significantly lower in the diabetic (D) group when compared to the obese (O) (*p* = 0.001) and to the obese and diabetic (OD) (*p* = 0.001) groups. Body fat was significantly lower in the empagliflozin-treated group (ODE) when compared to the obese and diabetic (OD) group (*p* = 0.002). There were no significant differences between the other groups.

Visceral fat was significantly lower in the control group (C) when compared to the obese (O) (*p* < 0.001), to the obese and diabetic (OD) (*p* < 0.001), and to the empagliflozin-treated group (ODE) (*p* < 0.001). It was also significantly lower in the diabetic (D) group when compared to the obese (O) (*p* < 0.001), to the obese and diabetic (OD) (*p* < 0.001), and to the empagliflozin-treated group (*p* < 0.001). Visceral fat was significantly lower in the empagliflozin-treated group (ODE) when compared to the obese and diabetic (OD) group (*p* < 0.014). There were no significant differences between the other groups (Table 1).

### 2.2. Hemoglobin Levels Were Significantly Higher among the Empagliflozin Treated Patients

Hemoglobin levels were significantly higher in the empagliflozin-treated group (ODE) when compared to the diabetic (D), and obese and diabetic (OD) groups (*p* = 0.004 and *p* < 0.001, respectively). There was no significant difference between the diabetic (D) and the obese and diabetic (OD) group (*p* = 0.850). The obese group (O) had a significantly higher hemoglobin when compared to the obese and diabetic group (OD) and a significantly lower level when compared to the empagliflozin-treated obese group (ODE) (*p* = 0.033 and *p* = 0.007 respectively) (Table 2).

### 2.3. Renal Parameters Were Significantly Higher in Diabetic Patients, Yet Were Reduced in the Empagliflozin Treated Group

Urea nitrogen level increases significantly with the appearance of diabetes in obesity (O vs. OD) (*p* = 0.002). In the empagliflozin-treated group (ODE), the urea nitrogen level was significantly lower when compared to the obese and diabetic (OD) group (*p* = 0.008) (Table 2).

Creatinine significantly increases with the appearance of diabetes in the obese groups (O vs. OD) (*p* = 0.011). In the empagliflozin-treated group (ODE), the creatinine level was significantly lower when compared to the obese and diabetic (OD) group (*p* = 0.012) (Table 2).

### 2.4. Blood Glucose and HbA1c Levels Were Significantly Higher in Diabetic Patients, Yet There Was No Significant Difference between the Different Diabetic Groups

Blood glucose and HbA1c levels were significantly lower in the control group (C) when compared with the diabetic (D), the obese and diabetic (OD), and the empagliflozin-treated obese and diabetic groups (*p* = 0.029, *p* = 0.005, and *p* < 0.001, respectively). Blood glucose and HbA1c levels were significantly lower in the obese group when compared with the diabetic (D), the obese and diabetic (OD), and the empagliflozin-treated obese and diabetic groups (*p* = 0.015, *p* = 0.008, and *p* < 0.001, respectively). There were no significant differences between the other groups regarding blood glucose and HbA1c levels.

### 2.5. Leptin Levels Were Significantly Higher in Obese Patients, Yet Were Reduced in the Empagliflozin-Treated Group

Leptin levels were significantly higher with the appearance of obesity (O) (*p* = 0.003) even if obesity was present with diabetes (OD) (*p* < 0.001) when compared to the control (C) group. It was also significantly higher in diabetic patients (D) when compared with the control group (C) (*p* = 0.029). Obese and diabetic patients (OD) had a significantly higher level of leptin when compared to diabetic yet not obese (D) patients (*p* = 0.001). In the empagliflozin-treated group (ODE), the leptin level was significantly lower when compared to the obese and diabetic (OD) group (*p* = 0.048) (Table 2).

### 2.6. There Were No Significant Differences between the Other Measured Parameters

There was no significant difference in body muscle percentage, white blood cell count, red blood cell count, platelet count, fibrinogen levels, uric acid, triglyceride, sodium and potassium levels, and thyroid-stimulating hormone levels among the groups.

There was no significant difference in the cholesterol levels among the groups. It bears mentioning, cholesterol levels were strongly affected by the antihyperlipidemic agents.

The continuous variables did not differ from the normal distribution. Data are shown as means ± standard deviation.

## 3. Discussion

In our clinical study, we examined metabolic and inflammatory parameters, kidney function, and leptin levels among patients afflicted with hypertension, obesity, type 2 diabetes, and cardiovascular diseases. The aim of our study was to detect the severity of the metabolic state among these patients and to examine a subgroup of patients treated with empagliflozin. In our study, we found empagliflozin-treated obese, diabetic patients had significantly lower BMI, body fat, and visceral fat values as well as lower serum creatinine and leptin levels when compared to patients with obesity and type 2 diabetes treated with usual antidiabetics (such as biguanides and sulfonylureas). Leptin levels were already higher among patients with type 2 diabetes even with normal BMI, and were significantly higher in obese non-diabetic patients and were the highest in obese patients with type 2 diabetes. Furthermore, we discovered that increased visceral fat and leptin levels predicted diabetes similarly to HbA1c.

Excess visceral adiposity is a major risk factor for metabolic and cardiovascular disorders. It plays a crucial role in the development of a diabetogenic and atherogenic metabolic profile inducing insulin resistance and increased cardiometabolic risk [26]. In our study, BMI, body fat, and visceral fat percentage were the highest among patients with obesity and type 2 diabetes (Group OD). In the empagliflozin-treated obese, diabetic patients (Group ODE), BMI, body fat, and visceral fat were significantly lower when compared with obese and diabetic patients (OD) treated with usual antidiabetics (Table 1). In an animal study, empagliflozin suppressed weight gain by shifting energy metabolism towards fat utilization, elevated adenosine monophosphate-activated [AMP] protein kinase, and acetyl coenzyme A [acetyl-CoA] carboxylase phosphorylation in skeletal muscle. Furthermore, empagliflozin increased energy expenditure, heat production and browning, and attenuated obesity-induced inflammation and insulin resistance by polarizing M2 macrophages in white adipose tissue [WAT] and liver [27]. Thus, empagliflozin suppressed weight gain by enhancing fat utilization and browning and attenuated obesity-induced inflammation and insulin resistance.

White adipose tissue is an endocrine organ capable of producing and releasing numerous bioactive substances known as adipokines or adipocytokines. Dysregulated production of adipocytokines is involved in the development of obesity-related diseases. Leptin is one of the most examined adipokines. An increased leptin level is associated with insulin resistance and T2DM development [28]. In T2DM, a link has also been reported between high leptin concentrations and increased cardiovascular [CV risk], including the presence of microvascular complications and cardiac autonomic dysfunction [29]. Furthermore, obesity, hypertension, metabolic syndrome, and endothelial dysfunction are more frequent in T2DM patients with increased leptin levels [30]. In chronic heart disease (CHD) patients, elevated leptin levels were significantly associated with an increased risk of cardiac death, acute coronary syndrome, non-fatal MI, stroke, and hospitalization for congestive heart failure [31,32]. Similarly, higher leptin levels were significantly related to the number of stenotic coronary arteries and arterial stiffness in CHD patients [33]. The presence, severity, extent, and lesion complexity of coronary atherosclerosis have been associated with higher leptin levels in CHD patients [34]. Leptin may also affect cardiac remodeling, metabolism, and contractile function [35]. Other effects of leptin include activation of inflammatory responses, oxidative stress, thrombosis, and atherosclerosis, thereby resulting in endothelial dysfunction and atherosclerotic plaque [16].

In our study, the leptin level was already higher among patients with type 2 diabetes even with normal BMI (Group D), was significantly higher in obese non-diabetic patients (Group O), and was the highest in obese patients with type 2 diabetes (Group OD) when compared to the control group.

A link between increased plasma leptin concentrations and chronic kidney disease (CKD) has been reported, which is possibly due to reduced renal clearance [36]. Leptin concentrations gradually increased with the severity of CKD [37]. In CKD patients, plasma leptin levels have been inversely associated with glomerular filtration rate and directly associated with urinary albumin levels as well as age and obesity markers (BMI and waist circumference) [38]. Overall, hyperleptinemia has been linked to the presence, severity, and progression of CKD. In our study, creatinine levels were significantly higher with the appearance of diabetes and were the highest among obese patients with type 2 diabetes. Among the empagliflozin-treated obese and type 2 diabetic patients, the creatinine level was significantly lower eliciting improved renal function (Table 2).

We possess a vast amount of knowledge regarding the cardiovascular and renal effects of SGLT2 inhibitors [20,39,40,41,42]. In addition to their direct effect on glucose homeostasis, they have many other underlying mechanisms from which not all are fully understood. For instance, SGLT2 inhibitors may also act upon visceral adipose tissue. Dapagliflozin therapy was associated with a decreased circulating leptin level and an increased circulating adiponectin level among patients with type 2 diabetes, which, may contribute to the beneficial effects of SGLT2 inhibitors on metabolic homeostasis, such as improved insulin resistance and reduced cardiovascular risk [43,44,45]. Furthermore, dapagliflozin displayed significantly lower arterial stiffness in diabetic mice treated with dapagliflozin when compared to untreated diabetic mice [46]. The effects of empagliflozin on adipocytokines were examined in an animal study conducted on obese rats. Empagliflozin dose-dependently reduced body weight, body fat, adiponectin, and leptin following the 28-day treatment [39]. In our study, the leptin level was significantly lower in the empagliflozin-treated obese and type 2 diabetic patients (ODE) when compared to the obese, diabetic patients (OD) treated with other antidiabetics (Table 2). To the best of our knowledge, this is the first time the beneficial effect of empagliflozin on the leptin level has been demonstrated in a clinical setting.

HbA1c is a well-known screening and diagnostic tool in detecting diabetes. A score higher than 5.7 % value implies prediabetes, and consequently, higher than 6.5 % confirms diabetes. Our receiver operating characteristic [ROC] analysis has proven the recommended 5.7 % cut-off value effectively predicted altered glucose homeostasis with very high sensitivity and acceptable specificity. In the same analysis, leptin was found to be similar in the prediction of diabetes. This is congruent with previous observations stating elevated leptin levels are associated with insulin resistance and T2DM development [28].

The second ROC analysis with the composite endpoint diabetes and obesity showed, in addition to HgA1c, leptin, and visceral fat may have a role in the diagnosis of diabetes among obese adults. These findings emphasize patients with increased visceral fat, which is easily measured using a smart weight scale, are prime candidates to be screened for insulin resistance or diabetes with HbA1c and fasting glucose value.

Hemoglobin values were the highest in the empagliflozin-treated group, which, may imply a slight hemoconcentration, and may be related to the osmotic diuretic effect of empagliflozin treatment. It is worthwhile to draw the attention of patients to the need for adequate fluid intake during SGLT2 inhibitor treatment. Unexpectedly, HbA1c levels were the highest in the empagliflozin-treated group. Presumably, this is due to the fact that, in Hungary, SGLT2 inhibitor treatment can only be prescribed to patients with an HbA1c level above 7%. This also means this group is a more severe patient group in terms of diabetes, thus, the results obtained prove even more crucial.

There was no significant difference in C-reactive protein (CRP) levels among the examined groups; however, some differences were detected. The CRP level was the lowest in the non-obese, non-diabetic group (C). Although many factors can influence the CRP level, it may be important that it was higher among obese and diabetic patients, which may indicate a low level of inflammation and corresponds to previous observations [19]. Among patients receiving empagliflozin treatment (ODE), the CRP level was lower when compared to the obese and diabetic group (OD), which may reflect lower inflammation status, likely due to the empagliflozin treatment. It has been previously reported, that empagliflozin reduced renal inflammation and oxidative stress in spontaneously hypertensive rats [47] In the EMPA-CARD trial patients with type2 diabetes and coronary artery disease treated with empagliflozin had lower levels of interleukin 6, interleukin 1*β* and CRP levels compared to a placebo. There were elevations in superoxidase dismutase (SOD) activity, glutathione (GSHr), and total antioxidant capacity (TAC) with empagliflozin [48].

Notably, there was no significant difference in LDL cholesterol levels. This may be due to the fact in which LDL cholesterol levels were greatly influenced by antihyperlipidemic drugs. Previous literature data indicated a moderate increase in LDL level can be detected with SGLT2 inhibitor treatment. In our study, we did not observe higher LDL values in the empagliflozin-treated group when compared to the other groups. Additionally, in our study, CV disease incidence was provided primarily to describe the patient population. Although it was lower in the empagliflozin-treated group, it was not intended to examine this correlation.

The main strength of our study is that, to the best of our knowledge, this is the first examination that has demonstrated that empagliflozin treatment has a beneficial effect on serum leptin levels under clinical conditions. However, our study was conducted on a relatively small number of patients, so further studies on a larger patient population are needed to confirm our results.

## 4. Materials and Methods

### 4.1. Ethics

The study protocol was approved by the Regional Ethics Committee of Pecs (No. 7622—PTE 2019) and was conducted in accordance with the ethical principles stated in the Declaration of Helsinki. Written informed consent was obtained from all patients.

### 4.2. Patients

102 patients (35 female, 67 male) were enrolled in our study. Patients were recruited from different internal medicine and outpatient departments by various physicians. They voluntarily agreed to participate in our study in which they signed an informed consent letter. Subgroup analysis was performed based on different metabolic states. Patients who did not have type 2 diabetes and were not obese were assigned to group C (20 patients), declared as the control group. Obese patients without diabetes were assigned to group O (obese), (20 patients). Non-obese patients with type 2 diabetes were selected into group D (diabetic), (19 patients). Obese and diabetic patients were assigned into group OD (obese and diabetic), (19 patients). Obese, diabetic patients receiving empagliflozin therapy for at least 3 months were assigned to group ODE (20 patients). Patients were considered obese if their BMI was 30.0 kg/m^2^ or higher. Antihypertensive, antidiabetic, and antihyperlipidemic therapies were recorded from the patient’s history as well as their comorbidities, such as diabetes mellitus, hypertension, and cardiovascular diseases. Exclusion criteria include the following: previous SGLT2 inhibitor therapy for groups C, O, D, OD; active cancer disease; and refusing to sign the consent form. Four patients were excluded from the study for different reasons (low compliance, severe epileptic seizure, withdrawal of their consent, and urgent psychiatric ward admission).

Patients’ general characteristics were as follows. The mean age for different groups was: 65.95 for group C, 66.40 for group O, 74.58 for group D, 70.90 for group OD, and 65.20 for group ODE. The distribution of sex (male to female percentage) in the groups was as follows: 75–25% for group C, 50–50% for group O, 52.60–47.40% for group D, 68.40–31.60% for group OD, and 75–25% for group ODE. Mean BMI values for different groups were as follows: 26.01 kg/m^2^ for group C, 34.75 kg/m^2^ for group O, 26.50 kg/m^2^ for group D, 35.78 kg/m^2^ for group OD, and 31.61 kg/m^2^ for group ODE. All patients had high blood pressure in their medical history. All patients in the diabetic groups (D, OD, ODE) had identified type 2 diabetes mellitus in their medical history, whereas none were reported in the remaining groups (C, O). The percentage of patients with identified cardiovascular disease was 70.40% in group C, 69.40% in group O, 78.60% in group D, 64.30% in group OD, and 73.48% in group ODE. All patients received antihypertensive therapy. All diabetic patients (D, OD, ODE) received antidiabetic therapy, whereas none were administered in the non-diabetic groups (C, O). Empagliflozin was administered only in the ODE group. No other SGLT2 inhibitors were used in our study. The percentage of patients with antihyperlipidemic therapy was as follows: 70% for group C, 75% for group O, 89.47% for group D, 84.20% for group OD, and 80% for group ODE.

### 4.3. Study Design

102 patients were recruited into this clinical study. We assessed their body composition, followed by pre-prandial venous blood collected using a peripheral venous catheter in the cubital vein. The preparation and laboratory procedures were in full accordance with the recommendations of the laboratory kits. Laboratory tests were performed at the Department of Laboratory Medicine, University of Pecs, Pecs, Hungary. The leptin levels were determined using the immunoassay method (Human Leptin ELISA, Biovendor, Czech Republic) at the Department of Biochemistry and Medical Chemistry, University of Pecs, Pecs, Hungary.

### 4.4. Anthropometric Measurements

The patients’ body composition was assessed using an Omron HBF-511 body composition scale (Omron HealthCare Co., Ltd., Kyoto, Japan). We measured weight, BMI, body fat percentage, and visceral fat percentage. Height was measured using a measuring tape.

### 4.5. Laboratory Tests

Pre-prandial laboratory tests were performed on every patient. These include complete blood count (red and white blood cell count, platelet count, hemoglobin level, hematocrit), fibrinogen, basic metabolic panel (pre-prandial glucose, sodium, potassium, calcium, blood urea nitrogen, and creatinine levels), lipid panel (total cholesterol, HDL cholesterol, LDL cholesterol, and triglyceride levels), liver panel (aspartate transaminase (AST), alanine transaminase (ALT), gamma-glutamyl transferase (GGT) levels), hemoglobin A1C level, and the thyroid stimulating hormone level.

### 4.6. Immunoassay Tests

Plasma leptin 1 levels were measured in duplicate using enzyme-linked immunosorbent assay (ELISA) kits (Cat. No. RD191001100). The blood samples were centrifuged at 2500× *g* for 10 min. The recovered plasma was stored at −70 °C in aliquots until assayed. The tests were performed in full accordance with the recommendations of the manufacturer, with a detection limit of 0.08 and 0.2 ng/mL, respectively. (BioVendor GmbH., Brno, Czech Republic).

### 4.7. Statistical Analysis

IBM SPSS statistics, version 28.0.0. (SPSS, Chicago, IL, USA, 2022); software for statistical; was used to conduct descriptive analyses and to describe the sample. Data are shown as means ± standard deviation.

Differences in the continuous variables were evaluated using a one-way repeated ANOVA statistical test (Tamhane post-hoc test) following the administering of the Kolmogorov–Smirnov test to check the normality of the data distribution. The continuous variables did not differ from the normal distribution.

In the case of categorical variables, data are shown as percentages and incidence (absolute number compared to total number). Differences were evaluated by using chi-square test analyses.

Multivariate linear regression and stepwise analyses of the data were performed regarding the leptin values for HbA1c, LDL, triglyceride, creatinine, hemoglobin, and visceral fat.

Multiple regression analysis with various models including leptin, HbA1c, and visceral fat considering the principle of multicollinearity was performed to reveal which factors predict the occurrence of diabetes and obesity.

The diagnostic power of variables was assessed using the area under the curve (AUC) of the receiver operating characteristic (ROC) curve. The predicted probabilities were calculated from the variables produced by binary logistic regression analysis, in which *p* ≤ 0.05 was considered statistically significant.

## 5. Conclusions

BMI, body fat, and visceral fat values as well as serum creatinine and leptin levels were improved with empagliflozin treatment. High leptin levels and leptin resistance in obesity are associated with insulin resistance, type 2 diabetes, increased risk of CV diseases, low-grade inflammation, and thrombosis. The markedly decreased circulating leptin levels observed in the empagliflozin-treated group may contribute to the known beneficial cardiovascular effects of empagliflozin treatment.

## Figures and Tables

**Table 1 ijms-24-04405-t001:** Patients’ general characteristics, n = number of patients, BMI = body mass index, kg = kilogram, m^2^ = square meter, Body fat = body fat percentage, Visc. fat = visceral fat, HT = high blood pressure, DM = type 2 diabetes mellitus, CVD = cardiovascular disease, SGLT2i = Sodium glucose co-transporter 2 inhibitors.

Group	C (n = 20)	O (n = 20)	D (n = 19)	OD (n = 19)	ODE (n = 20)	Total (n = 98)
Demographics and anthropometrics						
Age, years	65.95 ± 1.98	66.40 ± 2.23	74.58 ± 6.38	70.90 ± 1.74	65.2 ± 1.86	68.52 ± 0.90
Male sex, %	75.00	50.00	52.60	68.40	75.00	64.30
BMI, kg/m^2^	26.01 ± 0.50	34.75 ± 0.85	26.50 ± 0.44	35.78 ± 0.91	31.61 ± 0.77	31.04 ± 0.52
Body fat, %	26.75 ± 6.73	38.44 ± 8.38	28.12 ± 6.62	37.24 ± 6.67	30.98 ± 6.06	32.93 ± 6.90
Visc. fat, %)	10.5 ± 0.56	16.65 ± 0.93	10.89 ± 0.58	19.01 ± 1.25	15.50 ± 0.67	14.51 ± 0.79
Comorbidities						
HT, %	100.00	100.00	100.00	100.0	100.00	100.00
DM, %	0.00	0.00	100.00	100.00	100.00	59.20
CVD, %	70.40	69.40	78.60	84.70	64.30	73.48
Medications						
Antihypertensive, %	100.00	100.00	100.00	100.00	100.00	100.00
Antidiabetics (other, than SGLT2i), %	0.00	0.00	100.00	100.00	100.00	59.20
Antihyperlipidemic, %	70.00	75.00	89.47	84.20	80.00	79.59

**Table 2 ijms-24-04405-t002:** Laboratory parameters in the different groups. C = control group, n = number of patients, O = obese group, D = diabetic group, OD = obese diabetic group, ODE = obese diabetic group treated with empagliflozin, g = gram, L = liter, mmol = millimole, mL = milliliter, CRP = C-reactive protein, mg = milligram, µmol = micromole, Total chol. = total cholesterol, HDL = High-density lipoprotein cholesterol, LDL = Low-density lipoprotein cholesterol, ng = nanogram.

Groups	C (n = 20)	O (n = 20)	D (n = 19)	OD (n = 19)	ODE (n = 20)
Hemoglobin, g/L	141.85 ± 20.16	139.85 ± 11.85	133.79 ± 18.20	126.32 ± 14.72	152.90 ± 10.56
HbA1c, %	5.48 ± 0.08	5.67 ± 0.93	6.72 ± 0.34	6.39 ± 0.15	7.68 ± 0.33
Blood glucose, mmol/L	5.48 ± 0.85	5.62 ± 1.17	6.79 ± 1.95	6.20 ± 1.53	7.01 ± 1.61
CRP, mg/L	1.93 ± 0.43	6.81 ± 2.08	3.83 ± 1.08	4.55 ± 1.61	3.94 ± 0.60
Urea nitrogen, mmol/L	6.30 ± 0.75	5.22 ± 0.32	6.34 ± 0.42	9.69 ± 0.28	5.71 ± 0.31
Creatinine, µmol/L	83 ± 3.99	81.55 ± 3.42	94.68 ± 3.75	120.26 ± 9.75	81.71 ± 3.87
Total chol., mmol/L	4.73 ± 0.25	4.44 ± 0.26	3.79 ± 0.24	4.12 ± 0.36	4.12 ± 0.28
HDL, mmol/L	1.33 ± 0.08	1.25 ± 0.06	1.11 ± 0.05	1.10 ± 0.08	1.06 ± 0.05
LDL, mmol/L	3.15 ± 0.25	2.58 ± 0.21	2.21 ± 0.22	2.38 ± 0.31	2.21 ± 0.23
Triglycerides, mmol/L	1.76 ± 0.38	1.77 ± 0.27	1.81 ± 0.19	1.78 ± 0.22	1.88 ± 0.16
Leptin, ng/mL	5.97 ± 0.70	19.42 ± 3.06	10.33 ± 2.21	29.86 ± 3.61	17.43 ± 2.99

## Data Availability

The data presented in this study are available on request from the corresponding author. The data are not publicly available due to ethical and privacy restrictions.

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
