# Peer review of "Clinical Study of Metabolic Parameters, Leptin and the SGLT2 Inhibitor Empagliflozin among Patients with Obesity and Type 2 Diabetes"

_ijms, 2023, doi:10.3390/ijms24054405_

Round 1

Reviewer 1 Report

In this paper, the authors sought to investigate the metabolic state and leptin level among patients with obesity and type 2 diabetes mellitus, and the effect of empagliflozin on these parameters.

To this end, the authors recruited 102 patients into their clinical study, then they performed anthropometric, laboratory and immunoassay tests. Body mass index, body fat, visceral fat, urea nitrogen, creatinine and leptin levels were significantly lower in the empagliflozin treated group compared to the obese and diabetic patients receiving conventional antidiabetic treatments. Interestingly, leptin was increased not only in obese patients but in type 2 diabetic patients as well. Body mass index, body fat and visceral fat percentages were lower, and renal function was preserved in patients receiving empagliflozin treatment. In addition to the known beneficial effects of empagliflozin on the cardio-metabolic and renal systems, it may also influence leptin resistance.

Overall, this is an interesting paper on a topic of great interest. I congratulate the authors on this nice study.

Author Response

Answer to Reviewer 1

Dear Reviewer,

Thank you very much indeed for your time reviewing our manuscript! We are grateful for your positive feedback.

Reviewer 2 Report

1. Why authors have not measured (or maybe measured but not presented) the blood glucose levels in the patients in their study. This is relevant and important info to have as patients are being treated with a novel antidiabetic treatment.

2. It would be useful to include the information about what usual antidiabetics (beside empagliflozin) patients were being treated with”. It is not clear whether the empagliflozin-treated OD patients (ODE group) have been taking antidiabetics other than empagliflozin before or during the study? What about the patients in D group? If patients in any group have been treated with empagliflozin and conventional antidiabetics, do authors think there might be potentially a synergistic effect on the measured parameters. This can be discussed in the discussion.

3. It would be useful if authors discuss why they think empagliflozin-treated patients have lower body fat and visceral fat? What is the mechanism shown by others?

4. Patients in ODE group have higher levels of HgbA1c than OD group or D group. This is an interesting and somewhat unexpected finding, considering patients are being treated with an antidiabetic compound, empagliflozin. In this regard, authors discussed this briefly in line 224-231. Does this mean patients included in ODE group, had to have HgbA1c levels more than 7% to be included receive empagliflozin? If that was the case, it order to determine whether empagliflozin had any effect on HbA1c in ODE patients, the level of HbA1c should be presented historically. For example, HbA1c levels for at least 3 months prior can be presented to show whether 3 (or more) month of treatment with empagliflozin had any effect on HbA1c levels in the same patients compared to before they start empagliflozin. This should be possible since according to the authors, the inclusion criteria for ODE patients was that they had to receive empagliflozin for at least 3 months.

5. Has the beneficial effect of empagliflozin on CRP levels been reported in the animal or other clinical studies before? If yes, authors can give some references, as this is an interesting observation.

Note:  Figure 1 and 2 were missing in the version I received for review, but authors were referring to these figures. Only Table 1 and 2 were included!

Author Response

Answer to Reviewer 2

Dear Reviewer,

Thank you very much indeed for your time and efforts reviewing our manuscript! We are grateful for your useful observations and remarks. We did our best answering your questions and hope that the revised version will be eligible for you.

Why authors have not measured (or maybe measured but not presented) the blood glucose levels in the patients in their study. This is relevant and important info to have as patients are being treated with a novel antidiabetic treatment.

Answer:

Thank you for the suggestion! We have included blood glucose levels in the manuscript and in Table 2. (Page 5)

2.4. Blood glucose and HbA1c levels were significantly higher in diabetic patients, yet there was no significant difference between the different diabetic groups

            Blood glucose and HbA1c levels were significantly lower in the control group (C) when compared with the diabetic (D), the obese and diabetic (OD) and the empagliflozin treated obese and diabetic groups (p = 0.029, p = 0.005 and p < 0.001 respectively). Blood glucose and HbA1c levels were significantly lower in the obese group when compared with the diabetic (D), the obese and diabetic (OD) and the empagliflozin treated obese and diabetic groups (p = 0.015, p = 0.008 and p < 0.001 respectively). There were no significant differences between the other groups regarding the blood glucose and HbA1c levels. (Page 4, 130-139)

It would be useful to include the information about “what usual antidiabetics (beside empagliflozin) patients were being treated with”. It is not clear whether the empagliflozin-treated OD patients (ODE group) have been taking antidiabetics other than empagliflozin before or during the study? What about the patients in D group? If patients in any group have been treated with empagliflozin and conventional antidiabetics, do authors think there might be potentially a synergistic effect on the measured parameters. This can be discussed in the discussion.

Answer:

Thank you for the suggestion! We have included more details regarding the usual antidiabetic therapies in our patients:

In our study, we have found that empagliflozin treated obese, diabetic patients had significantly lower BMI, body fat and visceral fat values as well as lower serum creatinine and leptin levels when compared to patients with obesity and type 2 diabetes treated with usual antidiabetics (such as biguanides and sulfonylureas).

(Page 5, 172)

It would be useful if authors discuss why they think empagliflozin-treated patients have lower body fat and visceral fat? What is the mechanism shown by others?

Answer:

We added more details about the association between empagliflozin therapy and body composition:

In an animal study empagliflozin suppressed weight gain by shifting energy metabolism towards fat utilization, elevated AMP-activated protein kinase and acetyl-CoA carboxylase phosphorylation in skeletal muscle. Furthermore, empagliflozin increased energy expenditure, heat production and browning, and attenuated obesity-induced inflammation and insulin resistance by polarizing M2 macrophages in WAT and liver [27]. Thus, empagliflozin suppressed weight gain by enhancing fat utilization and browning and attenuated obesity-induced inflammation and insulin resistance.

(Page 5, 183-186, Page 6, 187-190)

Patients in ODE group have higher levels of HgbA1c than OD group or D group. This is an interesting and somewhat unexpected finding, considering patients are being treated with an antidiabetic compound, empagliflozin. In this regard, authors discussed this briefly in line 224-231. Does this mean patients included in ODE group, had to have HgbA1c levels more than 7% to be included receive empagliflozin? If that was the case, it order to determine whether empagliflozin had any effect on HbA1c in ODE patients, the level of HbA1c should be presented historically. For example, HbA1c levels for at least 3 months prior can be presented to show whether 3 (or more) month of treatment with empagliflozin had any effect on HbA1c levels in the same patients compared to before they start empagliflozin. This should be possible since according to the authors, the inclusion criteria for ODE patients was that they had to receive empagliflozin for at least 3 months.

Answer:

Presumably the HbA1c and blood glucose levels were higher prior to the SGLT2i therapy, which is supported by the data that was available for us. Unfortunately, we cannot access all the patients’ previous laboratory result prior the SLGT2i therapy, since the data of different medical providers were not always available for us.

Has the beneficial effect of empagliflozin on CRP levels been reported in the animal or other clinical studies before? If yes, authors can give some references, as this is an interesting observation.

Answer:

We added more details regarding the association between CRP levels and empagliflozin therapy:

It has been previously reported, that empagliflozin reduced renal inflammation and oxidative stress in spontaneously hypertensive rats [47] In the EMPA-CARD trial patients with type2 diabetes and coronary artery disease treated with empagliflozin had lower levels of interleukin 6, interleukin 1β and CRP levels compared to placebo. There were elevations in superoxidase dismutase (SOD) activity, glutathione (GSHr), and total antioxidant capacity (TAC) with empagliflozin [48].

(Page 7, 267-272)

Note:  Figure 1 and 2 were missing in the version I received for review, but authors were referring to these figures. Only Table 1 and 2 were included!

Answer:

The Editor decided to remove the mentioned figures, thus we have removed the references to these figures from the manuscript.

Reviewer 3 Report

I have received to review the review article entitled “Clinical study of metabolic parameters, leptin and the SGLT2 inhibitor empagliflozin among patients with obesity and type 2 diabetes”, prepared by Zsolt Szekeres et al. submitted to the International Journal of Molecular Sciences (IF=6.208). Obesity and the metabolic syndrome and its components are a very important challenge for public health today. They contribute to the development of type 2 diabetes, which is one of the most important risk factors for the development of cardiovascular diseases. Cardiovascular diseases are the leading cause of morbidity and mortality worldwide. Therefore, the topic discussed in this article is up-to-date and extremely important. The paper is generally well prepared and in my opinion it should be considered for publication but I would like to suggest some recommendations which can further improve the quality and attractiveness of the manuscript.

1)     Although the introduction is quite informative, it would be worth to add some information. The Authors mentioned oxidative stress and its role in the pathogenesis of cardiometabolic disease. In the population of obese patients there are metabolically healthy and metabolically unhealthy people. It should be mentioned that presence of metabolic disturbances in obese patients is associated with significantly increased oxidative stress. Moreover, it should be written that obesity and insulin resistance is the component of the metabolic syndrome most strongly associated with oxidative stress. Oxidative stress take a part in modification of lipoproteins which can make them more atherogenic. Nitrated lipoproteins have been recently discussed to take a part in the development of cardiovascular dysfunction in patients with diabetes. (10.1155/2021/9987352; 10.3390/antiox11010079; 10.1007/s13679-020-00375-0; 10.3389/fendo.2020.00027.)

2)     In the description of the statistical analysis methodology, the Authors wrote that for quantitative variables, the results were presented using the mean and standard deviation. In my opinion, such measures of central tendency and dispersion are adequate for continuous variables whose distributions do not differ significantly from the normal distribution. For variables that differ significantly from the normal distribution, the median and the interquartile range should be used. Please refer to this. So, were all quantitative variables normally distributed? If not, why did the Authors not decide to use the median and interquartile range for variables with distributions significantly different from the normal distribution?

3)     Strengths and limitations of the study should be described thoroughly in the discussion.

4)     I suggest to use the abbreviation HbA1c instead of HgbA1c.

5)     Although the work is written understandably, it should be checked by a specialist in terms of language.

6)     The reference list is generally well prepared, but minor adjustments are needed. For example, the commas between the abbreviation of the journal name and the year of publication should be removed, as they are not present in the method of citing references adopted in this journal.

Author Response

Answer to Reviewer 3

Dear Reviewer,

Thank you very much indeed for your time and efforts reviewing our manuscript! We are grateful for your useful observations and remarks. We did our best answering your questions and hope that the revised version will be eligible for you.

Although the introduction is quite informative, it would be worth to add some information. The Authors mentioned oxidative stress and its role in the pathogenesis of cardiometabolic disease. In the population of obese patients there are metabolically healthy and metabolically unhealthy people. It should be mentioned that presence of metabolic disturbances in obese patients is associated with significantly increased oxidative stress. Moreover, it should be written that obesity and insulin resistance is the component of the metabolic syndrome most strongly associated with oxidative stress. Oxidative stress take a part in modification of lipoproteins which can make them more atherogenic. Nitrated lipoproteins have been recently discussed to take a part in the development of cardiovascular dysfunction in patients with diabetes. (10.1155/2021/9987352; 10.3390/antiox11010079; 10.1007/s13679-020-00375-0; 10.3389/fendo.2020.00027.)

Answer:

Thank you for the suggestion! We have included a more detailed discussion regarding oxidative stress and its role in cardiometabolic disorders as well:

The presence of metabolic disturbances in obese patients results in oxidative stress [22]. Since obesity and insulin resistance is a major component of the metabolic syndrome, it is strongly associated with oxidative stress [23]. The oxidative modification of lipoproteins can result in more atherogenic compounds, which may have a key role in the development of cardiovascular dysfunction in patients with diabetes mellitus [24, 25].

(Page 2, 73-78)

In the description of the statistical analysis methodology, the Authors wrote that for quantitative variables, the results were presented using the mean and standard deviation. In my opinion, such measures of central tendency and dispersion are adequate for continuous variables whose distributions do not differ significantly from the normal distribution. For variables that differ significantly from the normal distribution, the median and the interquartile range should be used. Please refer to this. So, were all quantitative variables normally distributed? If not, why did the Authors not decide to use the median and interquartile range for variables with distributions significantly different from the normal distribution?

Answer:

Thank you for the suggestion! We have included more details regarding our statistical analysis and data variables:

The continuous variables did not differ from the normal distribution.  (Page 4, 156; Page 9, 359-360)

Strengths and limitations of the study should be described thoroughly in the discussion.

Answer:

We added more details about our investigation’s strengths and limitations in the text:

The main strength of our study is that, to the best of our knowledge, this is the first examination that has demonstrated that empagliflozin treatment has a beneficial effect on serum leptin levels under clinical conditions. However, our study was conducted on a relatively small number of patients, so further studies on a larger patient population are needed to confirm our results.

(Page 7, 281-285)

I suggest to use the abbreviation HbA1c instead of HgbA1c.

Answer:

Thank you for your suggestion! We have corrected the abbreviations.

Although the work is written understandably, it should be checked by a specialist in terms of language.

Answer:

Thank you for your suggestion! The manuscript has been checked by a specialist in terms of language (Jon Marquette, English language lector, Department of Languages for Biomedical Purposes and Communication)

The reference list is generally well prepared, but minor adjustments are needed. For example, the commas between the abbreviation of the journal name and the year of publication should be removed, as they are not present in the method of citing references adopted in this journal.

Answer:

The reference list has been adjusted according to the journal’s citing method.

Round 2

Reviewer 3 Report

The paper has been significantly improved. I recommend it for publication in its current form.